# Information-Theoretic Measures of Metacognitive Efficiency: Empirical Validation with the Face Matching Task

**DOI:** 10.3390/e27040353

**Published:** 2025-03-28

**Authors:** Daniel Fitousi

**Affiliations:** Department of Psychology, Ariel University, Ariel 40700, Israel; danielfi@ariel.ac.il

**Keywords:** metacognition, confidence, metacognitive efficiency, mutual information, KL divergence, Jeffrey’ divergence

## Abstract

The ability of participants to monitor the correctness of their own decisions by rating their confidence is a form of metacognition. This introspective act is crucial for many aspects of cognition, including perception, memory, learning, emotion regulation, and social interaction. Researchers assess the quality of confidence ratings according to *bias*, *sensitivity*, and *efficiency*. To do so, they deploy quantities such as meta−d’-$d^{\prime }$ or the M−ratio These measures compute the expected accuracy level of performance in the primary task (Type 1) from the secondary confidence rating task (Type 2). However, these measures have several limitations. For example, they are based on unwarranted parametric assumptions, and they fall short of accommodating the granularity of confidence ratings. Two recent papers by Dayan and by Fitousi have proposed information-theoretic measures of metacognitive efficiency that can address some of these problems. Dayan suggested meta−I and Fitousi proposed meta−U, meta−KL, and meta−J. These authors demonstrated the convergence of their measures on the notion of metacognitive efficiency using simulations, but did not apply their measures to real empirical data. The present study set to test the construct validity of these measures in a concrete behavioral task—the face-matching task. The results supported the viability of these novel indexes of metacognitive efficiency, and provide substantial empirical evidence for their convergence. The results also adduce considerable evidence that participants in the face-matching task acquire valuable metaknowledge about the correctness of their own decisions in the task.

## 1. Introduction

Assessing the validity of our own decisions is a self-reflective act of immense survival value [1,2,3,4]. For example, while learning for a test, we have to constantly monitor the level of our knowledge. If we feel that a satisfactory level of knowledge has been accomplished, we can halt our preparation; otherwise, we have to continue our learning [5,6]. This self-reflective ability is a form of *metacognition* [7,8,9,10,11,12,13]. Metacognition plays a central role in various domains, including learning [14], memory [15,16], self-awareness [17], action [18,19], psychiatric disorders [20], and social interaction [21]. Several neural substrates are involved in metacognitive assessments and monitoring of behavior [22]. These include the right anterior dorsolateral prefrontal cortex and the bilateral parahippocampal cortex [11]. A prominent method of assessing metacognition behaviorally is based on collecting confidence judgments from participants [13]. Measuring the quality of these confidence ratings is consequential for understanding the underlying mechanisms of metacognition. The goals of the present study are threefold: (a) to introduce several recently developed information-theoretic measures of metacognitive efficiency that are based on confidence judgments [3,4], (b) to demonstrate how these measures can be applied to a concrete behavioral task of face matching [23], and (c) to show that these information-theoretic measures converge on a single theoretical construct in this empirical context [24]. Previous studies by Dayan [3] and Fitousi [4] demonstrated the viability of these measures by using synthetic data from simulations. However, these measures should be computed and tested with real data from genuine empirical tasks. If indeed those information-based measures gauge the same psychological construct – metacognitive efficiency – they should correlate with other canonical measures, as well as with themselves, in real data.

A primary objective of research on metacognition is that of measuring the quality of confidence judgments. Three quantities stand out as crucial aspects of measurement and modeling: *bias*, *sensitivity*, and *efficiency*. To motivate our understanding of these concepts, consider a simple task by which an observer is asked to detect whether a signal is present or absent by emitting one of two possible responses, and then immediately the observer is asked to rate their confidence in the correctness of their decision. It is often customary to call the first task a *type 1* task, and the second a *type 2* task [25,26]. It is also theoretically convenient to separate the actor, who performs in the type 1 task, from the rater, who engages in the type 2 task, although the two entities might be the same person [27]. Sensitivity plays a role in both the actor’s and the rater’s decisions. The actor’s sensitivity is reflected in her ability to separate between noise and noise plus signal. In the language of signal detection theory (SDT, [28,29]), this quantity is captured by the parameter $d^{\prime }$. The rater’s sensitivity reflects their ability to distinguish between the actor’s correct and incorrect decisions. A metacognitively sensitive rater tends to provide high confidence ratings for the actor’s correct decisions, and low confidence ratings for the actor’s incorrect decisions. However, the sensitivity of the rater cannot serve as a viable measure of the rater’s quality of decision. Trivially, a rater can be highly sensitive if the actor is giving correct decisions most of the time. In other words, the rater’s job becomes much easier as the actor becomes more sensitive. Thus, a genuine measure of metacognitive efficiency should correct the rater’s sensitivity for the actor’s performance. Moreover, metacognitive efficiency can be affected by bias. An overconfident rater, one who gives high confidence ratings even when the actor’s decisions are incorrect, is effectively both insensitive and inefficient.

Model-free [30] and model-based [1,2,26,27,31] approaches have been developed to quantify metacognitive bias, sensitivity, and efficiency. One of the most influential model-based frameworks in the field is called the $meta-d^{\prime }$ model [1,2]. The model relies on principles of SDT [28,29], and as such, incorporates parametric assumptions [25,26,32]. The model’s objective is to predict the rater’s hypothetical sensitivity in the type 1 task based on the rater’s performance in the type 2 task. The value of this theoretical quantity is dubbed $meta-d^{\prime }$. Figure 1A–D illustrates the theoretical and empirical components of this model and their interrelations. Figure 1A presents the underlying model of the type 1 task. This is the traditional equal-variance signal detection model [29], depicted by two overlapping Gaussian distributions, one for the noise and one for the signal + noise. The normalized distance between the two distributions is captured by the parameter $d^{\prime }$, which gauges the sensitivity of the actor in the type 1 task. Figure 1B illustrates the underlying model behind the rater’s type 2 confidence judgments. It assumes the rater’s decisions are also based on an equal-variance signal detection process. However, the Gaussian distributions reflect the incorrect and correct actor’s responses, and are generated from the type 1 distributions by adding noise. Figure 1D depicts the hypothetical Receiver Operating Characteristic (ROC) curve of the rater’s type 2 performance. The parameters of the theoretical models can be found by fitting them to the data of the accuracy–confidence histograms that are illustrated in Figure 1C. These histograms reflect the empirical normalized probabilities of the rater’s confidence ratings conditioned on the accuracy of the actor’s response. The distributions for the correct responses appear in green, whereas the distributions for the incorrect responses appear in red. The probabilities in each distribution sum up to 1.

$Meta-d^{\prime }$ reflects the expected $d^{\prime }$ of the rater in the type 1 task based on her performance in the type 2 task. Because this quantity and the actor’s type 1 $d^{\prime }$ are expressed in the same measurement units [2], they can be directly compared. Metacognitive efficiency can then be assessed in several possible ways: as an M-ratio ($meta-d^{\prime }/d^{\prime }$) or as a difference (meta−d’-$d^{\prime }$). The rater’s efficiency is optimal when M−ratio=1, or when meta−d’-$d^{\prime }=0$. The rater is hyposensitive when M−ratio<1 or when meta−d’-$d^{\prime }<0$. There are situations in which the rater is hypersensitive, namely when M−ratio>1, or when meta−d’-$d^{\prime }>0$ [1]. This can occur when the rater has access to additional information accumulated after the actor’s decision. Such a scenario can be modeled, for example, by the second-order model [27]. According to this model, the actor and the rater are sampling evidence from two separate distributions. Hypersensitivity in this model can be modeled by reducing the noise (i.e., variance) in the rater’s distribution to a much lower level than the noise (i.e., variance) in the actor’s distribution

Measures emanating from the $meta-d^{\prime }$ model [1,2], such as meta−d’-$d^{\prime }$ and the M−ratio, have been used extensively by researchers to evaluate the quality of confidence judgments [34]. However, these measures are not without their limitations [3]. First, it is not immediately clear why empirical confidence ratings should necessarily accommodate to a type 1 decision process [3]. Moreover, these measures require distributional assumptions, are not bias-free, are applicable only to ratio-scale variables, do not scale with the number of stimuli, responses, or confidence rating alternatives, do not have definite upper and lower bounds, and are applicable only to experimental designs with two stimuli and two responses. Dayan [3] and Fitousi [4] have recently proposed information-theoretic indexes of metacognitive efficiency that can ameliorate most of these problems, and serve as additional means of assessment.

Dayan [3] has succinctly summarized the utility of an information-theoretic approach to judgments of confidence: “the problem of confidence is inherently one of information—that the actor has about the true state of the stimulus; and that the rater has about the same quantity and about what the actor used. It therefore seems appropriate to use the methods of information theory to judge the relationship between the stimulus, the actor and the rater” (p. 11). Indeed, the information-theoretic measures have many desirable characteristics: they are relative measures, bias-free (e.g., do not depend on criterion shifts), do not require any parametric assumptions, scale with the number of confidence rating points, do not require the assumption of ratio or order of confidence ratings, and are relatively easy to compute. Both Dayan [3] and Fitousi [4] performed simulation with canonical models to support the validity of the information-theoretic measures they developed. Both authors demonstrated convergence of their information-theoretic measures based on synthetic data. However, neither of them demonstrated the viability of these measures with real empirical datasets, nor did they examine whether the construct validity of those measures was achieved in a genuine empirical setting. The present study aims to address these objectives.

## 2. An Information-Theoretic Approach to Metacognitive Efficiency

Recent years have seen a renewed interest [35] in information-theoretic approaches to perception and cognition [36,37,38,39,40,41,42,43,44]. Many researchers have begun to realize the potential strength of information theory [45,46] in providing powerful tools for addressing the uncertainty that characterizes brain and behavior [47,48]. In two recent studies, Dayan [3] and Fitousi [4] have developed complementary information-theoretic approaches to metacognition that build on the idea of confidence judgments as a communication system. An illustration of such a system is presented as a Venn diagram in Figure 2. This Venn diagram depicts a communication model that consists of three entities: Stimuli, an Actor, and a Rater. The uncertainty associated with each variable is denoted by a circle in a different color. This diagram can guide rudimentary insights into the informational quantities that affect the actor’s and rater’s accuracy. For example, the discriminability of the actor’s response can be quantified by the mutual information shared between the stimuli and the actor. The rater’s accuracy depends on the amount of mutual information they share with the actor.

### 2.1. Meta−I

Dayan [3] outlined an information-theoretic measure of metacognitive efficiency he called meta−I. Consider an actor who performs in a two-choice signal detection task with consequences of a = 1, or a = −1. The actor’s response is *r*, and they are correct when r=a and incorrect when r≠a. The confidence–accuracy discrete distributions can be used to assess the metacognitive efficiency of the rater as a difference between two entropies.(1)$$\begin{array}{ccc} meta-I & = & H_{2}\left(r|a=-1\right)-H_{2}\left(r|c,a=-1\right) \end{array}$$
where(2)$$\begin{array}{ccc} H_{2}\left(r|a=-1\right) & = & h_{2}\left[P\left(r|a=-1\right)\right] \end{array}$$
and(3)$$\begin{array}{ccc} H_{2}\left(r|c,a=-1\right) & = & \sum_{c}P\left(c|a=-1\right)h_{2}\left[P\left(r|c,a=-1\right)\right] \end{array}$$
where(4)$$\begin{array}{ccc} h_{2}\left[P\left(x\right)\right] & = & \sum_{x=0;1}-P\left(x\right)\log_{2}P\left(x\right) \end{array}$$
is the entropy of a Bernoulli random variable. The first entropy $H_{2}\left(r|a=-1\right)$ is the overall uncertainty about the actor’s accuracy. In the SDT model it is determined by the value of the parameter $d^{\prime }$. When the actor is perfectly accurate and $d^{\prime }\to \infty$ or when the actor is perfectly wrong, this entropy amounts to 0 bits; when the actor is completely uncorrelated with $d^{\prime }$, which occurs when $d^{\prime }\to 0$, this entropy amounts to 1 bits. The second entropy $H_{2}\left(r|c,a=-1\right)$ is the weighted average uncertainty about the residual accuracy after observing the confidence rating *c* where the weights arrive from the probability of observing that rating *c*.

The meta−I measure has many desirable characteristics: it is a relative measure, does not require any parametric assumptions, is bias-free, scales with the number of confidence rating points, does not require the assumption of ratio or order of confidence ratings, and offers definite bounds from above and from below.

The meta−I is an ingenious measure in its own right. However, it can be modified to become a more general index of metacognitive efficiency. Note that the measure is contingent on accuracy (i.e., it depends on a specific partition, r = a, r≠ a), while there is evidence that participants do not necessarily rate confidence based on the accuracy of response [49]. For example, Koriat [50,51] has demonstrated the operation of a *consensuality principle* in judgments of confidence, whereby participants give the highest confidence rating to the most popular option, not necessarily to the one that is objectively correct. Moreover, meta−I was originally developed to assess metacognitive efficiency in the two-choice SDT task (i.e., it assumes two mutual choices, a = 1 or a = −1). However, meta−I does not apply to tasks of multi-alternative choice [49]. For example, assume an experimental design with four stimuli that are categorized into four responses, and are then rated on a four-point confidence scale. In that case, meta−I is not readily applicable. Thus, a more general version of meta−I, which I call $meta-I^{\prime }$, is needed. This version is not contingent on accuracy, and makes no assumptions about the structure of the experimental design (i.e., number of stimuli or/and responses). The new $meta-I^{\prime }$ proposed here meets these requirements. It is computed as the mutual information between response and the joint confidence-stimuli variable:(5)$$\begin{array}{cc} meta-I^{\prime }=I\left(r;c,s\right) & =H(r|s)-H(r|s,c)\\ \; & =H(r)+H(c,s)-H(s,r,c) \end{array}$$This quantity amounts to 0 bits when the actor shares no information with the stimuli and rater; it is maximal when the actor shares all the information with the stimuli and the rater. An upper-bound on $meta-I^{\prime }$ is determined by the variable with the lowest number of alternatives. For example, if the stimuli variable has the lowest number of alternatives, say four, then the upper bound is 2 bits. In the case of a two-choice SDT task, the upper bound is 1 bit. Decomposition of $meta-I^{\prime }$ into its entropy terms can provide more insights. According to one possible decomposition, it reflects the degree to which uncertainty in the actor’s response to the stimuli H(r|s) is reduced after observing the rater’s confidence judgments H(r|s,c). According to another decomposition, $meta-I^{\prime }$ reflects the reduction in the combined uncertainty in the actor’s response and the confidence-stimuli variables H(r)+H(s,c) after removing the uncertainty in the joint stimuli-response-confidence variable H(s,r,c). In its new form, $meta-I^{\prime }$ can be applied to complex experimental designs, with several stimuli and responses, and without the need to define the correctness of the response.

### 2.2. Meta−U, Meta−KL, and Meta−J

Fitousi [4] has recently developed three complementary information-theoretic measures of metacognitive efficiency, which he called meta−U, meta−KL, and meta−J. The first measure bears close affinity with Dayan’s meta−I, and its generalized form $meta-I^{\prime }$. The measure quantifies metacognitive efficiency as the mutual information exchanged between the rater’s confidence judgments and the stimuli-response variable:(6)$$\begin{array}{cc} meta-U=I(c;s,r) & =H(c|s)-H(c|s,r)\\ \; & =H(c)+H(s,r)-H(s,r,c) \end{array}$$The index expresses the amount of information gained about confidence from knowing the actor’s response, after knowing the stimuli. When the rater’s metacognitive efficiency is null, meta−U amounts to zero bit, setting a lower bound on efficiency. An upper-bound on meta−U is placed by the variable with the lowest number of alternatives in the experiment. Decomposition of meta−U into entropy terms can explicate the logic of this measure. In the first decomposition, meta−U reflects the degree to which the conditional uncertainty of confidence after knowing the stimulus is reduced after observing the actor’s response. In the second decomposition, we note that meta−U gauges the degree to which uncertainty in the confidence and stimulus-response variables H(c)+H(s,r) is reduced after observing the actor’s response H(s,r,c). This measure corrects for type 1 performance by taking the difference between type 1 uncertainty H(s,r) and the type 1 and type 2 uncertainty H(s,r,c), in a similar way to meta−d’-$d^{\prime }$.

Importantly, it can be shown that meta−U equals $meta-I^{\prime }$ due to the symmetry of mutual information:(7)$$\begin{array}{c} meta-U=meta-I^{\prime }\\ I(c;s,r)=I(r;c,s) \end{array}$$In the case of a two-choice SDT task, meta−U and $meta-I^{\prime }$ are equal to Dayan’s meta−I.

The other two measures advanced by Fitousi [4], meta−KL and meta−J, are based on the *Kullback–Leibler* (KL) [52] and *Jeffrey’s* divergences [53]. These divergences are applied to the normalized confidence–accuracy distributions P[(confidence=y)|Correct)] and Q[(confidence=y)|Incorrect)]. It should be noted though that, in principle, these divergence measures can be derived for any required partition of the stimuli-response joint variable, irrespective of accuracy. Accuracy is defined by labels (i.e., “correct”, “incorrect”) placed by the experimenter on the actor’s responses. However, information-theoretic quantities are agnostic to such labels. A rater can be metacognitively efficient even when they flip the “correct” and “incorrect” labels on the actor’s responses.

The new measure proposed by Fitousi, meta−KL, is based on a KL divergence $D_{KL}\left(Q\|P\right)$ that measures the degree of separation between correct and incorrect accuracy–confidence distributions:(8)$$\begin{array}{ccc} meta-\mathrm{KL} & =D_{KL}\left(P\|Q\right) & =\sum P\left(c=y|Correct\right)\log_{2}\left(\frac{P(c=y|Correct)}{Q(c=y|Incorrect))}\right) \end{array}$$It amounts to 0 *bits* when the rater cannot distinguish between correct and incorrect responses of the actor, and increases as the rater becomes more metacognitively efficient. An upper bound is met when the two distributions are at their maximal separability. An analogous expression of meta−KL gauges the difference between the entropy in the joint correct and incorrect distribution and the entropy in the correct distribution:(9)$$\begin{array}{cc} meta-\mathrm{KL} & =D_{KL}\left(P\|Q\right)\\ \; & =H(P,Q)-H(P)\\ \; & =H(c=y|Correct,c=y|Incorrect)-H(c=y|Correct) \end{array}$$Metacognitive efficiency is null when uncertainty in the joint correct–incorrect confidence–accuracy distribution is equal to the uncertainty in the correct confidence–accuracy distribution. Metacognitive efficiency increases as the uncertainty in the correct distribution decreases. It reaches an upper bound when the latter is 0 bits; namely, the rater eliminates all uncertainty in the actor’s correct (incorrect) responses to the stimuli.

The meta−KL measure does not satisfy the triangle inequality, and is also not symmetric in the two distributions, since $D_{KL}\left(P\|Q\right)\neq D_{KL}\left(Q\|P\right)$. In this sense, meta−KL does not qualify as a metric [54]. To address these constraints, Fitousi [4] has also proposed meta−J based on the Jeffrey’s divergence [53,55], which offers a symmetric measure of divergence:(10)$$\begin{array}{cc} meta-J=J(P,Q) & =D_{KL}\left(P\|Q\right)+D_{KL}\left(Q\|P\right)\\ \; & =\sum P\left(c=y|Correct\right)\log_{2}\left(\frac{P(c=y|Correct)}{Q(c=y|Incorrect))}\right)+\\ \; & \sum Q\left(c=y|Incorrect\right)\log_{2}\left(\frac{Q(c=y|Incorrect)}{P(c=y|Correct))}\right) \end{array}$$This measure functions in a similar way to meta−KL. It equals 0 bits when the rater cannot distinguish between the correct and incorrect accuracy–confidence distributions, and increases as the rater becomes more efficient. However, unlike meta−KL, it takes into consideration the two possible KL divergences, $D_{KL}\left(P\|Q\right)$ and $D_{KL}\left(Q\|P\right)$, between the correct and incorrect confidence–accuracy distributions. Another way of expressing the same measure:(11)$$\begin{array}{cc} meta-J & =J(P,Q)\\ \; & =2\times H(P,Q)-H(P)-H(Q)\\ \; & =2\times H(c=y|Correct,c=y|Incorrect)-H(c=y|Correct)-H(c=y|Incorrect) \end{array}$$

suggests that metacognitive efficiency gauges the degree to which the rater reduces uncertainty in the actor’s response by eliminating uncertainty in the correct accuracy–confidence distribution, and/or in the incorrect accuracy–confidence distribution. The meta−J takes into consideration two sources of uncertainty, weighting them as two independent contributions to metacognitive efficiency. This is in contrast to the meta−KL, which gives all the weight to the uncertainty in the correct accuracy–confidence distribution.

To validate these novel measures, Fitousi [4] has performed simulations with the $meta-d^{\prime }$ model [1,33] using the free code published by Maniscalco and Lau [1]. He varied the values of $d^{\prime }$ and $meta-d^{\prime }$ from 0.01 to 4, incrementally in equal steps of 0.21 and in an orthogonal fashion. For each combination of $d^{\prime }$ and $meta-d^{\prime }$, he generated the corresponding confidence–accuracy distribution by simulating 10,000 trials. The Type 1 bias (the SDT parameter c) was set to 0. The number of ratings was fixed at 4. The information-theoretic measures meta−U, meta−KL, and meta−J were computed for each confidence–accuracy distribution, and plotted in three separate heatmaps, with $d^{\prime }$ in the abscissa and $meta-d^{\prime }$ in the ordinate. The code for the meta -d’ model was downloaded 10.08.2018 from: http://www.columbia.edu/ bsm2105/type2sdt/archive/index.html. The other codes can be found in https://osf.io/hsvfm/. Figure 3 presents the results of these simulations. As can be noted, all three heatmaps reveal one of the most important properties of an efficiency measures, namely that the rater’s sensitivity is corrected for Type 1 performance. To see why, note that the highest values of metacognitive efficiency are achieved when the actor’s sensitivity ($d^{\prime }$) is low and the rater’s sensitivity ($meta-d^{\prime }$) is high. In all three heatmaps, as the actor’s sensitivity increases, there is less and less room for the rater to accomplish high metacognitive efficiency. Also notable are the differences between the mutual information measure (meta−U), and the divergence measures (meta−KL, meta−J) with respect to the maximal values they can take and their overall dynamics.

The information-theoretic measures have many valuable characteristics. These are discussed in detail in the papers by Dayan [3] and Fitousi [4]. Here, I give a brief summary of the most important aspects: (a) relative measures, (b) bias-free, (c) do not require any parametric assumptions, (d) scale with the number of confidence rating points, (e) do not require the assumption of ratio or order of confidence ratings, (f) easy to compute, (g) offer definite lower and upper bounds on performance, and (h) can be related to other entropy measures at the neuronal level.

One important question that remains to be addressed is whether these measures of metacognitive efficiency converge on the same psychological construct. Previous efforts by Dayan [3] and Fitousi [4] have emulated various simulations with theoretical models to show that the measures capture the same psychological construct. However, there is still the need to assess the construct validity of these measures with real datasets. If these different indexes capture the same psychological construct, they should correlate in real empirical data. This hypothesis is subjected to scrutiny in the next section.

## 3. Empirical Validation

### 3.1. The Face-Matching Task

Here, I provide an empirical validation of the information-theoretic measures of metacognitive efficiency. To do so, I harness the face-matching task [56], which has become an important tool in assessing sundry aspects of face recognition [57,58,59,60]. In this task, participants are presented with two face images taken under various lighting conditions, with different cameras, and at different times. Participants are asked to decide whether a given pair of face images belong to the same identity, or alternatively to two different identities. Figure 4 presents examples of pairs of face images from the *Glasgow Face Matching Task* (GFMT, [56]) and the *Kent Face Matching Task* (KFMT, [61])—two of the most popular standard tests, which were also deployed in the present investigation. It turns out that with unfamiliar facial identities, the face-matching task poses considerable challenge for both humans [23,56,57,58,59,60] and machines [62]. This is mainly because under variable ambient conditions, images of the same personal identity can look completely different, whereas images of different identities may appear similar. Despite extensive research on the face-matching task [63,64], the underlying mechanisms are still not well understood. For example, many researchers believe that match and mismatch face pairs are governed by two separate mechanisms [58,59]. However, a recent paper by Fitousi [23] has challenged this position, by introducing an *unequal-variance signal detection model* (UVSD) of the face matching task. This model postulates a single unified mechanism that can account for performance with both match and mismatch stimuli. The model can also explain many related aspects of face matching, including Type 1 confidence judgments, similarity ratings, and the relations between Type 1’s accuracy and confidence.

### 3.2. Metacognition in the Face-Matching Task

Notwithstanding the wealth of research on the face-matching task [57,65,66,67], only little is known about the underlying metacognitive capabilities with which participants perform this task [68,69,70,71,72]. The majority of studies that investigated metacognition in face-recognition tasks [69,70,71,72] found medium or moderate levels of metacognitive abilities. These studies deployed face recognition tasks and questionnaire-based self-estimates. Other studies [73,74] showed that participants can provide confidence judgments that differentiate between correct and incorrect Type 1 performance. However, the measures deployed by those studies are suboptimal, mainly because they are subjected to the confounds mentioned earlier, namely bias and inability to correct for Type 1 performance. The application of the M−ratio index, along with the novel information-theoretic measures meta−U, meta−KL, and meta−J to the face-matching task, can ameliorate most of these shortcomings, and shed new light on the underlying mechanisms that govern face recognition. Most importantly, the joint application of these measures is needed to garner strong evidence for their construct validity and convergence.

The face-matching task is ideal for our present purposes. It is challenging enough to produce below-ceiling accuracy for all participants [23], enabling us to compute SDT and associated measures. Furthermore, it does not require an additional staircase procedure, which is known to inflate both model-based and model-free measures of metacognitive sensitivity [75]. Moreover, the task is implemented in various well-established face-matching tests [56,61], for which the normative type 1 performance characteristics are well known. A signal-detection model of the task has been recently advanced [23]. However, little is known about the metacognitive efficiency of observers in this task. Taken together, these factors make the face-matching task a compelling testing platform.

Two experiments were conducted. In Experiment 1, participants engaged in the long version of the GFMT [56]. In Experiment 2, a new sample of participants were tested with the short version of the KFMT [61].

## 4. Method

### 4.1. Participants

A total of 140 participants took part in Experiment 1 and Experiment 2. In total, 70 participants (14 men, 56 women, mean age = 23.03, sd = 2.14) participated in Experiment 1; and a second sample of 70 participants (16 men, 54 women, mean age = 22.7, sd = 3.1) participated in Experiment 2. The participants were recruited from the pool of participants of Ariel University. The study was approved by the Ariel University Ethical Committee (AU-SOC-DF-20210411). All participants gave their informed consent.

### 4.2. Stimuli and Apparatus

The set of stimuli in Experiment 1 consisted of the long version of the GFMT [56]. This includes 168 pairs of Caucasian, unfamiliar faces. The faces were high-quality grayscale photos taken in full-view with different cameras, conveying neutral expression and pose. Each image was approximately 12 × 7 cm. Half of the face pairs (84) were “identity-match”, and the other half were “identity-mismatch”. The set of stimuli in Experiment 2 consisted of the short version of the KFMT [61]. There were 40 pairs of unfamiliar faces, with each pair composed of two color photos: a frontal image on the right and a student ID image on the left. The frontal image was high-quality of approximately 283×332 pixels, with a neutral pose and expression. The student ID image was a color image of approximately 142×192 pixels. The ID images varied in pose, expression, and lighting. Half of the face pairs (20) were “identity-match”, and the other half were “identity-mismatch”.

### 4.3. Procedure and Design

The general method in both experiments was identical. Participants were sitting in front of a computer screen. In each trial, they were presented with a randomly selected face pair, and were asked to decide as accurately and as speedily as they can whether the pair constituted a “match” or a “mismatch” by pressing one of two computer keys. Immediately after responding, the images were removed from the screen, and a four-point confidence scale appeared on the screen with 1 = “not confident” and 4 = “highly confident”. In Experiment 1, participants completed two blocks of trials, which amounted to a total of 336 trials. In Experiment 2, participants completed four blocks, which resulted in a total of 160 trials. The experiments were run with the *macromedia Authorware 7* software.

## 5. Results

### 5.1. Face Matching

All data, codes, and supporting materials can be downloaded from https://osf.io/hsvfm/. The measures were computed for each participant in each experiment. Of the available parametric measures, I focus on the M−ratio measure, because it is considered to be the most reliable [76,77]. To assess the M−ratio, I fit the $meta-d^{\prime }$ and $d^{\prime }$ parameters to the data of each participant in each experiment using the code published by Maniscalco and Lau (downloaded 10 August 2018 from: http://www.columbia.edu/~bsm2105/type2sdt/archive/index.html). For five participants in each experiment, the resulting fitting procedure provided negative $meta-d^{\prime }$ values, so they were removed from the remaining analyses. The information-theoretic values do not require any fitting, and are computed directly from the empirical data using dedicated codes that I have developed and can be reproduced in https://osf.io/hsvfm/.

First, it would be valuable to look at the underlying individual differences in the Type 1 and Type 2 measures. Figure 5 and Figure 6 present the empirical frequency distributions of those variables in Experiment 1 and Experiment 2, respectively. These can provide insights into the nature of individual differences in both Type 1 and Type 2 performance, across the two experiments. These distributions can also teach us about the sampling characteristics of the parametric and non-parametric measures. One key observation is that the distributions are not necessarily symmetric. Those associated with the information-theoretic exhibit mostly skewed asymmetric negative distributions, with the majority of observations located on the lower end of the data range. The M−ratio shows this trend only in Experiment 2, but not in Experiment 1.

The GFMT administrated in Experiment 1 was relatively easier ($d^{\prime }$ = 2.51, sd = 0.65) than the KFMT in Experiment 2 ($d^{\prime }$ = 1.01, sd = 0.41), an outcome that is well-expected [23]. However, the results from both experiments pointed to the conclusion that, on average, participants have metaknowledge about their face-matching performance. The mean M−ratio amounted to 1.44 (sd = 0.66) in Experiment 1 and 2.83 (sd = 3.25) in Experiment 2. The M−ratio in Experiment 2 seems larger than often found in other domains [78]. This might be due to the relatively small number of trials in the short version of the KFMT. Recall that for M−ratio a value greater than 1 entails hypersensitivity. One-sided Student’s *t*-tests supported these conclusions, with all *t-statistics* being significant at the 0.05 level. The results of the M−ratio strongly suggest that participants in the face-matching task can monitor the level of their performance in the face-matching task, irrespective of their Type 1 level of performance.

Next, I turn to our novel information-theoretic measures: meta−U, meta−KL, and meta−J. To test whether participants acquired metaknowledge regarding their face-matching performance, I compared those values to 0 bits of information. The null hypothesis is that participants have no metaknowledge whatsoever, and the information-based measures equal 0 bits. The average meta−U was greater than 0 bit in both experiments. In Experiment 1, it amounted to 0.22 bits (sd = 0.17) [t(65)=12.45, p<0.001]. In Experiment 2, it was 20.37 bits (sd = 0.21) [t(65)=14.25, p<0.001]. Recall that in the two-choice task meta−U is quantitatively comparable to Dayan’s [3] meta−I. The average meta−KL value in Experiment 1 amounted to 1.54 bits (sd = 0.95) [t(65)=13.39, p<0.001]. In Experiment 2, it amounted to 1.11 bits (sd = 0.74) [t(65)=12.08, p<0.001]. The average meta−J in Experiment 1 amounted to 4.15 bits (sd = 2.37) [t(65)=14.43, p<0.001], and 2.54 bits (sd = 1.74) [t(65)=11.84, p<0.001] in Experiment 2. The information-theoretic measures are consistent with the M−ratio results, supporting the conclusion that participants in Experiment 1 and Experiment 2 acquired genuine metaknowledge about the correctness of their own decisions.

### 5.2. Testing for Construct Validity

A viable approach for testing *construct validity* [24] would be to see whether different measures of the same putative construct—metacognitive efficiency—correlate to some extent. I have, therefore, performed a series of linear regressions with all possible pairs of measures (M-ratio, meta−U, meta−KL, and meta−J), in order to quantify the corresponding *coefficient of determination* $R^{2}$, and *Pearson’s correlation coefficient r*. Figure 7 and Figure 8 present the results of these analyses, respectively, for Experiment 1 and Experiment 2.

As can be noted in Figure 7 and Figure 8, significant positive and strong relations ensued between each pair of measures in both experiments. Overall, higher $R^{2}$ values obtained for the KFMT (Experiment 2) than the GFMT (Experiment 1) data. In the former, all coefficient of determination values were larger than 0.47, or a Pearson correlation of r=0.68. In the latter, all those values exceeded 0.29, or a Pearson correlation of r=0.53, all significant at the p<0.05 level. The strongest relations were observed between meta−KL and meta−J ($R^{2}=0.94$). This is quite expected given their mathematical affinity. Taken together, these results confer validation support in the convergence of the information-theoretic measures on the common psychological structure of metacognitive efficiency. The convergence holds for both within-information measures and between SDT- and information-based measures. This strengthens our belief in their construct validity.

## 6. General Discussion

I have introduced several information-theoretic measures of metacognitive efficiency recently developed by Dayan [3] and Fitousi [4]. These indexes consist of the meta−I [3] and the $meta-I^{\prime }$, meta−U, meta−KL, and meta−J [4]. These measures were originally developed within a common theoretical framework, assuming a communication system with stimuli (experimenter), an actor, and a rater as encoders/decoders. I provided a succinct account of each measure, along with its possible interpretations. I then applied these information indexes, along with the M−ratio to data from two face-matching experiments [56]. The results provided considerable support in the convergence of these measures on a single psychological construct—metacognitive efficiency. The measures exhibited strong correlations both across-themselves and with the parametric M−ratio. Another important result that is particularly relevant to research on face recognition is that participants in the face-matching task exhibit significant metaknowledge about their performance in the task. This finding was supported by both the parametric measure M−ratio and the information-based measures.

One source of motivation for developing information-theoretic markers of metacognitive efficiency has to do with the limitations of existing model-based measures, such as $metad^{\prime }-d^{\prime }$ and M−ratio. These indexes are built on the dubious assumption that they should fit exactly to a type 1 decision process [3]. They are subjected to decisional bias [1,2]. They cannot accommodate changes in the granularity of type 2 confidence ratings [3], nor do they scale with the number of type 1 stimuli or responses [4]. Moreover, they cannot be applied in designs with more than two stimuli and two responses, namely in multi-attribute classification tasks [79]. Information-theoretic measures address most of these limitations. First, because these measures do not require parametric assumptions, they can be computed relatively easily. Second, because they do not assume a particular number of stimuli or responses, they can be applied to tasks which incorporate multi-alternative choices. Third, they can adapt to changes in the granularity of stimuli, response, and confidence alternatives. It is important to note that the deployment of information-based measures does not exclude the testing of parametric, hierarchical models [31,76]. As a matter of fact, these measures can serve as an important yardstick in the development of such models, mainly because they are expressed by the most general quantity of entropy, and as such, can transcend different assumptions concerning distributions, parameters, and psychological mechanisms. This fact is reflected in our present findings, which show that these measures are compatible with the M−ratio. Researchers can take advantage of the information-based measures to relate confidence at the behavioral and neuronal levels, since uncertainty at both levels can be quantified in bits. Another reason for advancing the information-theory approach to metacognition has to do with the suitability of such a theory to characterize capacity limitations [45], rate-distortions [80,81], information-bottlenecks [82,83], and other quantities that are not readily available in current approaches (e.g., SDT). Future work on confidence and metacognition should seek to expand our theoretical understanding by building on these powerful notions.

### 6.1. Dependency on Type 1 Performance

Applied metacognition researchers would ideally like to have a measure that captures the underlying construct of metacognitive efficiency in each individual without contamination by type 1 performance [76]. In practice, this ideal is rarely met. Using synthetic data from simulations with state-of-the-art parametric models, both Guggenmos [76] and Rausch and colleagues [77] have shown that the M-ratio measure is not completely independent of type 1 performance. Notably though, Guggenmos [76] found that, when tested with empirical data, the M-ratio exhibited only negligible dependency on type 1 performance, making it an almost ideal measure. Thus, a question that remains to be answered concerns the ability of information-theoretic measures to accommodate type 1 performance. This is an empirical question that can be tested. In any event, one may rightly argue that in some sense a parametric approach (e.g., meta - d’) is superior to the information-theoretic approach advanced here. This is primarily because the former allows us to know the underlying generating process, whereas the latter is apparently agnostic to this process. Indeed, several researchers [76,77,84,85] have argued that measures of metacognition should ideally be derived by capturing the hierarchical model of type 1 and 2 performance and making inferences on the critical parameters. If so, why should one advance information-theoretic measures in the first place? The answer to this question is threefold.

First, processing models of metacognition are difficult to develop for many cases. For example, at present, there are only a few process models for multi-alternative tasks [79]. The information-theoretic measures presented here can be readily applied to a wide scope of applications, including the multi-alternative case. Second, the deployment of information-theoretic measures does not necessarily exclude the development and testing of hierarchical parametric models [86]. On the contrary, because information-based measures are mostly invariant to the underlying parametric distributions, they can be computed and compared across a range of parametric models [86]. In this way, information-based measures can serve as a yardstick for model-building and testing. Third, entropy is ubiquitous across all system’s levels [42], including behavioral (reaction times, eye movements), neuronal (single-cell spikes, cell-ensemble signals), and chemical (ion concentration) levels. This domain-generality makes information-based measures incredibly instrumental in bridging remote levels of processing. For example, one can gain new insights by relating quantities of behavioral metacognitive efficiency with signatures of entropy generated by neuronal and chemical processing in the brain [41,86].

### 6.2. Practical Advice to Practitioners of Information-Based Measures

The convergence of the information-theoretic measures suggests that they can be used interchangeably. However, in practical applications, the meta−J, for example, is preferable to the meta−KL for two reasons. First, it is a metric, as it complies with the triangle equality [53]; second, it takes into consideration reduction in entropy from both the correct and incorrect accuracy–confidence distributions. The meta−KL measure, in contrast, is not a metric [54]; it also does not consider the reduction in entropy in the incorrect accuracy–confidence distribution. This is important if one is interested in dissociating two alternative mechanisms by which raters weight their evidence. This difference can account for *the positive evidence bias* [33,87,88,89,90], whereby raters overweight evidence in favor of the actor’s decision and underweight evidence against the actor’s decision. Moreover, the meta−U and $meta-I^{\prime }$ are of broader applicability than meta−KL and meta−J, mainly because they are not contingent on the accuracy of the actor’s response. As such, they can be applied to tasks in which the correctness of the actor’s response is not defined [91]. Another prominent advantage of all the information-theoretic measures is that they can be computed for tasks that require categorization of multi-attribute stimuli [49]. This is currently impossible to achieve with the meta−d’ approach. A related advantage of information-based measures is that type 1 performance can be measured by I(S;R)—the mutual information between stimuli and response [4]. In two-choice SDT designs, I(S;R) is monotonically related to $d^{\prime }$ [38]. This affords researchers to quantify both type 1 and type 2 performance in terms of information-theoretic quantities.

Another relevant issue that emerges from the application of information theory to metacognition concerns the granularity of the confidence judgment scale. If the raters use every point in their scale equally often, then a rater that operates with an eight-point scale can be more sensitive than a rater who operates with a four-point scale. The reason is that the former can transmit more information than the latter [92]. Both Dayan [3], and Fitousi [4] have shown via simulations that the information-theoretic measures accommodate the granularity of the confidence scale, that is, metacognitive efficiency can potentially improve as the number of rating alternatives increases. SDT-related measures do not exhibit this important characteristic. Moreover, the information-theoretic measures also accommodate the granularity of the stimuli and response alternatives. Finally, I would recommend researchers to deploy all three major information-theoretic measures together, if possible, to enhance the validity of their conclusions.

### 6.3. Implications for Face Recognition

The present effort bears important implications for research on face recognition and individual differences in metacognitive abilities. Do people have insights into their face-recognition capabilities? The majority of studies who tested for this question [69,70,71,72,73,74] found medium or moderate levels of metacognitive abilities. These studies employed self-reports, questionnaires, and confidence judgments. However, these tools are susceptible to the vagaries and biases mentioned at the outset, such as response bias, or differences in Type 1 performance. These problems are particularly acute in the face-recognition domain where the presence of large-scale individual differences in both Type 1 and Type 2 performance is well-known [93]. For example, on the one end of the spectrum, one can find participants with prosopagnosia [94]. These participant exhibit very poor face-recognition performance; while on the other end, there are *super-recognizers* [69]—participants with extraordinary face-recognition abilities. There are good reasons to believe that those two populations also differ in their metaknowledge concerning their type 1 performance. The upshot is that in order to study metacognitive abilities in the domain of face recognition, one must employ measures that correct for type 1 performance and for the possibility of response biases. The present study provides, what I believe to be, the first genuine evidence for high metacognitive sensitivity of observers in their face-recognition abilities. This evidence is not confounded by type 1 performance or bias. A related approach that can be tremendously instrumental in characterizing individual differences in both type 1 and type 2 performance is the development of hierarchical Bayesian models [31] of face-recognition tasks. Such models can assist in specifying the parameters of individual participants as resulting from higher hyper-parameters of the population. Although the information-theoretic measures are not parametric in nature, they can still be computed in such hierarchical models.

## Figures and Tables

**Figure 1 entropy-27-00353-f001:**
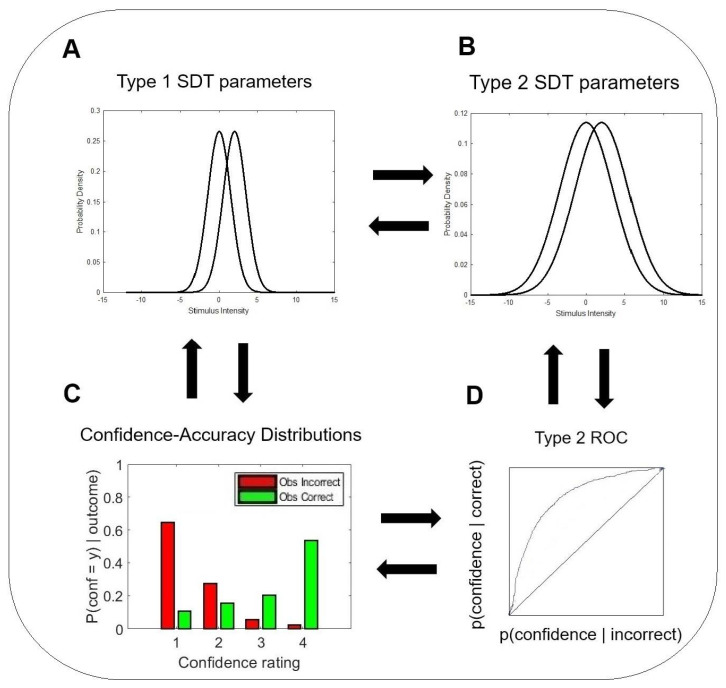
The meta - d’ model [1]. (**A**) A signal detection theory (SDT) representation of the actor’s decisions in the type 1 task. (**B**) An SDT representation of the rater’s decisions in the type 2 task. (**C**) Empirical confidence-rating distributions conditional on type 1 correct and incorrect responses. A rater with high metacognitive sensitivity provides high confidence ratings when the actor is correct, and low confidence ratings when the actor is incorrect. A rater with low metacognitive sensitivity does not distinguish between the actor’s correct and incorrect responses. (**D**) A theoretical ROC curve of the type 2 task is generated by assuming a hypothetical equal-variance SDT model of confidence. The meta - d’ model [1,31,33] expresses the type 2 hit rates p(Hit) and false alarms p(FA) in terms of the divergence units of the type 1 d’, and then looks for the sensitivity parameter that best fits the empirical confidence data. The resulting parameter is called $meta-d^{\prime }$.

**Figure 2 entropy-27-00353-f002:**
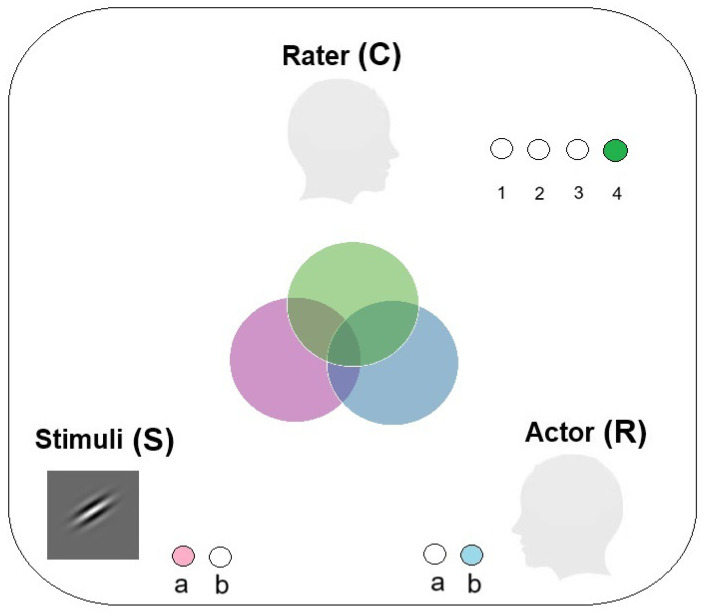
A Venn diagram illustrating a communication system between stimuli, actor, and rater. Uncertainty in Stimuli is depicted by a pink circle, uncertainty in an Actor’s response is illustrated by a blue circle, and uncertainty in a Rater’s confidence ratings is represented by a green circle. In this example, there are two stimuli. The actor responds by emitting one of two possible responses, and the rater expresses their confidence in the actor’s response by sending one of four confidence-rating levels.

**Figure 3 entropy-27-00353-f003:**
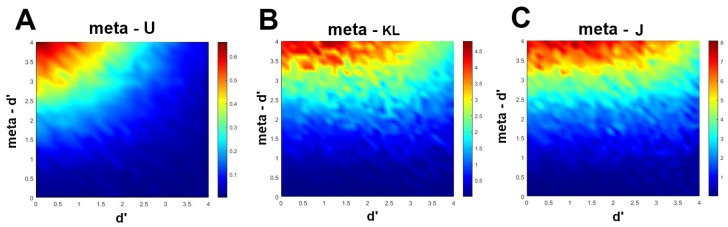
Theoretical validation of the information measures of metacognitive efficiency using the $meta-d^{\prime }$ model [1,33]. Data are generated by increasing incrementally and in an orthogonal fashion the values of parameters $d^{\prime }$ and $meta-d^{\prime }$, and then computing the corresponding values of meta−U, meta−KL, and meta−J. The resulting values are presented in a heatmap against the values of $d^{\prime }$ and $meta-d^{\prime }$. (**A**) meta−U. (**B**) meta−KL. (**C**) meta−J.

**Figure 4 entropy-27-00353-f004:**
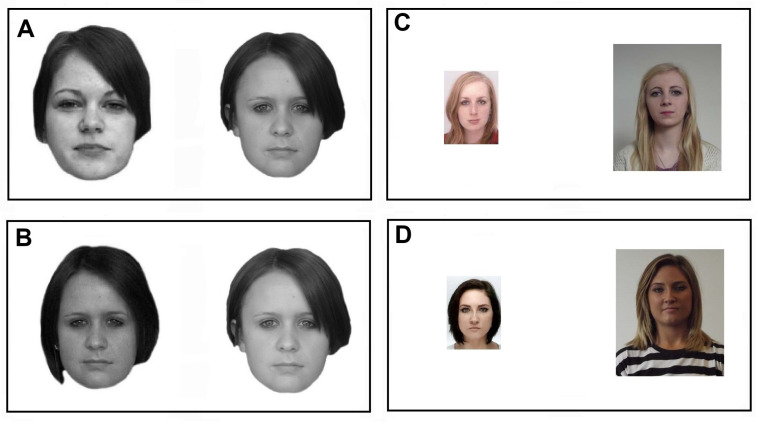
Examples of face pairs from the Glasgow Face Matching Task (GFMT, panels (**A**,**B**)) and from the Kent Face Matching Task (KFMT, panels (**C**,**D**)). Panels (**A**,**C**) depict the same identity (i.e., “match”), whereas panels B and D depict different identities (i.e., “mismatch”).

**Figure 5 entropy-27-00353-f005:**
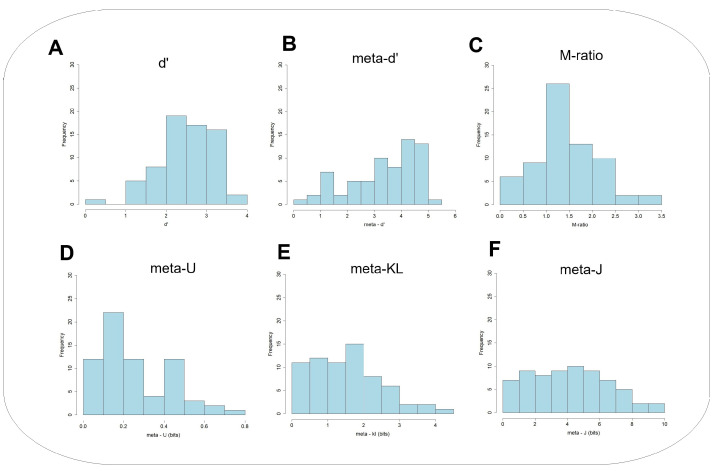
Experiment 1: Frequency distributions of the major quantities recorded in the GFMT (Glasgow Face Matching Task): (**A**) $d^{\prime }$, (**B**) $meta-d^{\prime }$, (**C**) M-ratio, (**D**) meta−U, (**E**) meta−KL, and (**F**) meta−J.

**Figure 6 entropy-27-00353-f006:**
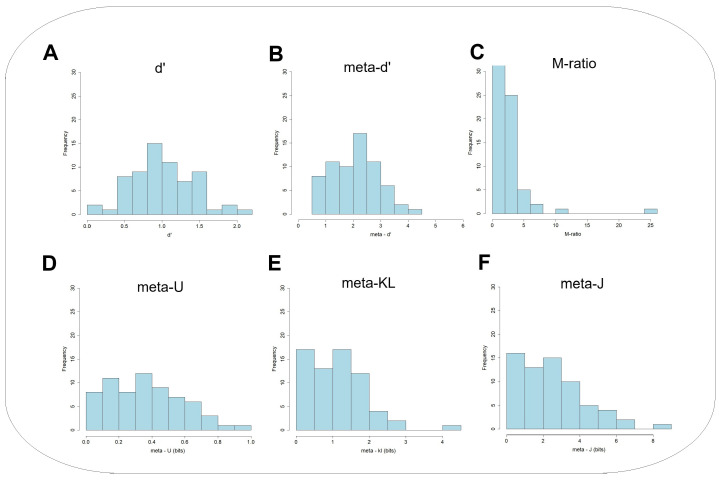
Experiment 2: Frequency distributions of the major quantities recorded in the KFMT (Kent Face Matching Task): (**A**) $d^{\prime }$, (**B**) meta - d’, (**C**) M-ratio, (**D**) meta−U, (**E**) meta−KL, and (**F**) meta−J.

**Figure 7 entropy-27-00353-f007:**
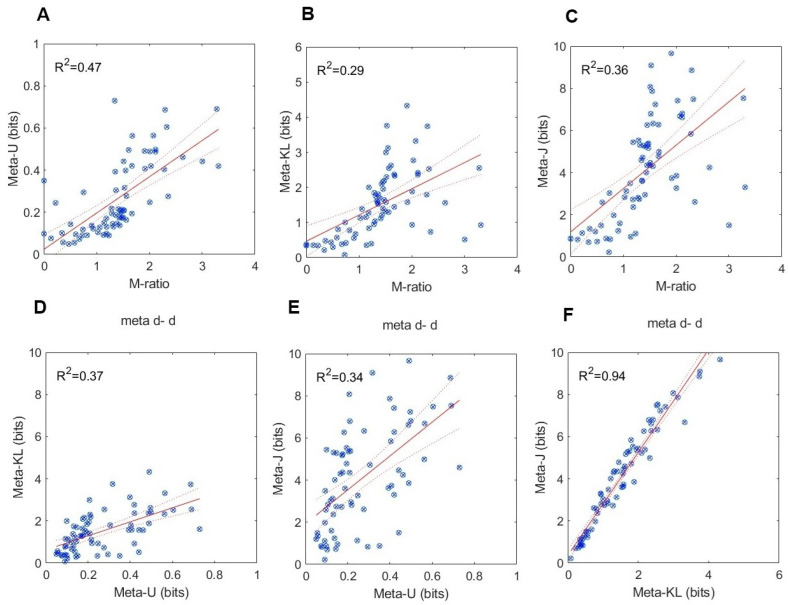
Experiment 1: Results from the GFMT (Glasgow Face Matching Task). (**A**–**C**) M-ratio as a function of meta−U, meta−KL, and meta−J. (**D**–**F**) Relations between meta−U, meta−KL, and meta−J. Blue circles are data points, solid red line is a regression line, and red dashed lines are the 95% confidence interval.

**Figure 8 entropy-27-00353-f008:**
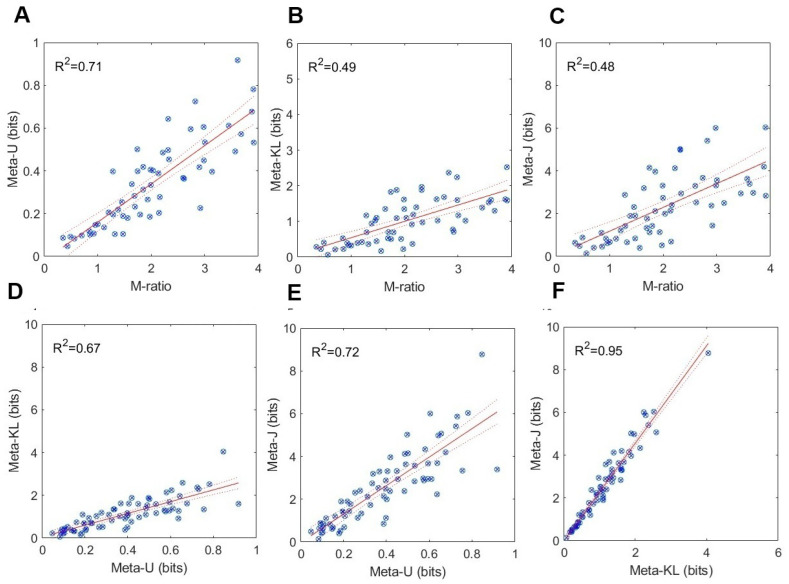
Experiment 2: Results from the KFMT (Kent Face Matching Task): (**A**–**C**) M-ratio as a function of meta−U, meta−KL, and meta−J. (**D**–**F**). relations between meta−U, meta−KL, and meta−J. Blue circles are data points, solid red line is a regression line, and red dashed lines are the 95% confidence interval.

## Data Availability

Data can be found in https://osf.io/hsvfm/.

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
