# Peer review of "Information-Theoretic Measures of Metacognitive Efficiency: Empirical Validation with the Face Matching Task"

_entropy, 2025, doi:10.3390/e27040353_

Round 1
Reviewer 1 Report
Comments and Suggestions for Authors
The present manuscript summarizes and extends recent attempts to quantify metacognitive efficiency in information-theoretic terms. In the first part, the author introduces and defines a set of entropy-based measures. In the second part, the author validates the measures based on empirical datasets. In sum, this is a timely and highly valuable contribution to ongoing discussions about proper ways to measure the construct of metacognitive ability, in particular the ability to gauge the correctness of one's own actions and perceptions.
Major comments:
1. My main issue with the article is that it advertises information-based measures as "accommodating to type 1 performance" and "not requir[ing] any parametric assumptions" without explaining the large caveats of both arguments. While I see that some of these phrases are carefully chosen, they likely will suggest to readers not deep into the matter that information-based measures solve the issue of type 1 performance and on top, accomplish this without any necessary parameter optimization.
To take one step back, what applied metacognition researchers *ideally would like to have* is a measure that captures the underlying construct of metacognitive efficiency in each individual without contamination by type 1 performance. This goal can only be achieved by capturing the hierarchical model of type 1 and 2 decision making and making inferences on the critical parameters. Descriptive measures -- such as entropy-based measures -- cannot achieve such inferences in an unbiased manner.
The careful wording in some places suggests that the author is aware of this, but I didn't see this discussed. Indeed, the author's Figure 3 would make this very point: simulating observers with different metacognitive ability levels (in this case based on the meta-d' model) yield varying estimates of the proposed entropy-measures across the type 1 performance dimension. Thus for any given value of e.g. meta-U there is ambiguity about the true underlying metacognitive ability that gave rise to the observed data. Obviously, the meta-d' model is only one of many possible models, and Figure 3 may look different for other models. But this just goes to show that without knowing and capturing the (approximately) true model, unbiased inferences are not guaranteed and likely not possible.
To put it strongly, this aspect *could* render a measure useless for a meaningful application in practice. The same goes for M-Ratio and other measures, but at least for M-ratio it has been shown that empirical M-ratios vary surprisingly little across datasets and observers with different type 1 performance levels (doi.org/10.1093/nc/niab040, Figure 3B). For entropy-based measures we don't know yet and hence at this point "accommodating to type 1 performance" -- which M-ratio and other measures do as well -- is hardly a delta argument in favour of entropy measures.
The fact that entropy-based measures are parameter-free is good for computational performance reasons, but I'm not sure it is otherwise an argument in their favour. The true underlying model is almost certainly hierarchical and parametric (if at the level of individual synaptic weights).
Overall, I would appreciate if these caveats are mentioned in the discussion and some wordings adjusted (e.g. "strong evidence suggests that these measures carry-over a residual dependency on type 1 performance" ... "information-theoretic indexes of metacognitive efficiency that can ameliorate most of these problems".).
2. If I'm not mistaken, entropy-based measures cannot capture hypersensitivity. Even worse, I would assume (though I may be wrong) that hypersensitivity artificially *reduces* the mutual information and hence hypersensitive observers would be converted to hyposensitive observers. Given the fact that this manuscript features datasets with strong apparent hypersensitivity (see 3.) increases the importance of this issue and it should be discussed or raised as a limitation.
3. The M-Ratios used for empirical validation (Figures 5 & 6) contain M-Ratios in a range that I have never seen before. For the Kent Face Matching Task, the fast majority of observers have M-Ratios exceeding 1 indicating strong to very strong hypersensitivity. I can imagine different technical reasons -- use of a staircase (doi.org/10.1093/nc/niz009), low performance (as d' is in denominator and -- given it's binary outcomes -- is typically noisy), or generally too little trials. If the nature of the task plausibly explains these extreme values, it should be explained. If not, my advice would be to provide a comparison with hierarchically estimated M-ratios (github.com/metacoglab/HMeta-d), possibly replacing meta-d'-d' (see 6.).
Minor comments:
4. I'm wondering which model the author used to generate data with the meta − d′ model. In particular, how exactly did the author vary "the values of [...] meta − d′ incrementally in small steps (p. 7)" and how did they compute confidence ratings from this? I realize that this sentence refers to another manuscript, but it would be important to know the metacognitive model underlying Figure 3. As a side note, it has only been recently shown that the underlying model of meta-d'/d' (i.e. M-ratio) is of a truncated Gaussian type (doi.org/10.1037/met0000634).
5. I think there is an error in Figure 6A in that it accidentally shows Meta-J and not Meta-U (Meta-J is also shown in Figure 6C).
6. I advice to not include analyses related to meta-d - d' in the figures (5D-F, 6D-F) and the text. The measure of meta-d - d' is conceptually flawed and thus only adds noise and no meaningful information. The flaw can be seen when e.g. considering an observer with very high metacognitive noise, such that meta-d' is essentially 0 irrespective of type 1 performance. In this case meta-d' ≈ d', and thus the strong type 1 performance difference of this measure becomes transparent (see also doi.org/10.1093/nc/niab040, Figures 2C & 3A). M-ratio does not contain this flaw, as meta-d' is in the nominator and observers with zero metacognitive introspection will have an M-ratio of 0 regardless of type 1 performance. Thus, I suggest focusing on M-ratio.
Signed
Matthias Guggenmos
Author Response
Reviewer 1
The present manuscript summarizes and extends recent attempts to quantify metacognitive efficiency in information-theoretic terms. In the first part, the author introduces and defines a set of entropy-based measures. In the second part, the author validates the measures based on empirical datasets. In sum, this is a timely and highly valuable contribution to ongoing discussions about proper ways to measure the construct of metacognitive ability, in particular the ability to gauge the correctness of one's own actions and perceptions.
I thank the Reviewer for their positive evaluation.
Major comments:
- My main issue with the article is that it advertises information-based measures as "accommodating to type 1 performance" and "not requir[ing] any parametric assumptions" without explaining the large caveats of both arguments. While I see that some of these phrases are carefully chosen, they likely will suggest to readers not deep into the matter that information-based measures solve the issue of type 1 performance and on top, accomplish this without any necessary parameter optimization.
I thank the Reviewer for their important comment. I agree with the Reviewer’s claim and in this version, I made a genuine effort to address this lacuna.
To take one step back, what applied metacognition researchers *ideally would like to have* is a measure that captures the underlying construct of metacognitive efficiency in each individual without contamination by type 1 performance. This goal can only be achieved by capturing the hierarchical model of type 1 and 2 decision making and making inferences on the critical parameters. Descriptive measures -- such as entropy-based measures -- cannot achieve such inferences in an unbiased manner.
I agree with the Reviewer that one cannot safely argue that entropy-based measures can achieve complete independence from type 1 performance.
The careful wording in some places suggests that the author is aware of this, but I didn't see this discussed. Indeed, the author's Figure 3 would make this very point: simulating observers with different metacognitive ability levels (in this case based on the meta-d' model) yield varying estimates of the proposed entropy-measures across the type 1 performance dimension. Thus for any given value of e.g. meta-U there is ambiguity about the true underlying metacognitive ability that gave rise to the observed data. Obviously, the meta-d' model is only one of many possible models, and Figure 3 may look different for other models. But this just goes to show that without knowing and capturing the (approximately) true model, unbiased inferences are not guaranteed and likely not possible.
I agree with the Reviewer.
To put it strongly, this aspect *could* render a measure useless for a meaningful application in practice. The same goes for M-Ratio and other measures, but at least for M-ratio it has been shown that empirical M-ratios vary surprisingly little across datasets and observers with different type 1 performance levels (doi.org/10.1093/nc/niab040, Figure 3B). For entropy-based measures we don't know yet and hence at this point "accommodating to type 1 performance" -- which M-ratio and other measures do as well -- is hardly a delta argument in favour of entropy measures.
I agree with the Reviewer.
The fact that entropy-based measures are parameter-free is good for computational performance reasons, but I'm not sure it is otherwise an argument in their favour. The true underlying model is almost certainly hierarchical and parametric (if at the level of individual synaptic weights).
I thank the Reviewer for this deep and thoughtful comment. The Reviewer is raising a very important issue. The Reviewer’s comment is taken to heart. I have now dedicated a paragraph to this issue and moderate the unwarranted claims.
Overall, I would appreciate if these caveats are mentioned in the discussion and some wordings adjusted (e.g. "strong evidence suggests that these measures carry-over a residual dependency on type 1 performance" ... "information-theoretic indexes of metacognitive efficiency that can ameliorate most of these problems".).
I thank the Reviewer for this proposal. I have now discussed these issues in the General Discussion. I have moderated the strong claims and removed the unwarranted claims. A specific section called “Dependency on Type 1 performance” is dedicated to discussing the Reviewer’s comments.
Applied metacognition researchers would ideally like to have a measure that captures the underlying construct of metacognitive efficiency in each individual without contamination by type 1 performance [69]. In practice, this ideal is rarely met. Using synthetic data from simulations with state-of-the-art parametric models, both Guggenmos [69] and Rausch and colleagues [72] have shown that the M-ratio measure is not completely independent of type 1 performance. Notably though, Guggenmos [69] found that, when tested with empirical data, M-ratio exhibited only negligible dependency on type 1 performance, making it a close-to-ideal measure. Thus, a question that remains to be answered concerns the ability of information-theoretic measures to accommodate type 1 performance. This is an empirical question that can be tested. In any event, one may rightly argue that in some sense a parametric approach (e.g., meta -d’) is superior to the information-theoretic approach advanced here. This is primarily because the former allows us to know the under- lying generating process, whereas the latter is apparently agnostic to this process. Indeed, several researchers [69,72–74] have argued that measures of metacognition should ideally be derived by capturing the hierarchical model of type 1 and 2 performance and making inferences on the critical parameters. If so, why should one develop information-theoretic measures in the first place? The answer to this question is threefold. First, process models of metacognition are difficult to develop for many cases. For example, at present there are only few process models for multi-alternative tasks [71]. The information-theoretic measures presented here can be readily applied to a wide scope of applications, including the multi-alternative case. Second, the deployment of information-theoretic measures does not necessarily exclude the development and testing of hierarchical parametric models [75]. On the contrary, because information-based measures are mostly invariant to the underlying parametric distributions, they can be computed and compared across a range of parametric models [75]. In this way, information-based measures can serve as a yardstick for model-building and testing. Third, entropy is ubiquitous across all system’s levels [42], including behavioral (reaction times, eye movements), neuronal (single-cell spikes, cell-ensemble signals), and chemical (Iion concentration) levels. This domain-generality makes information-based measures incredibly instrumental in bridging remote levels of processing. For example, one can gain new insights by relating quantities of behavioral metacognitive efficiency with signatures of entropy generated by neuronal and chemical processing in the brain [41,75].
- If I'm not mistaken, entropy-based measures cannot capture hypersensitivity. Even worse, I would assume (though I may be wrong) that hypersensitivity artificially *reduces* the mutual information and hence hypersensitive observers would be converted to hyposensitive observers. Given the fact that this manuscript features datasets with strong apparent hypersensitivity (see 3.) increases the importance of this issue and it should be discussed or raised as a limitation.
I thank the Reviewer for raising this issue. If I understand correctly the Reviewer brings the issue of whether the entropy-based measure can capture hypersensitivity. In fact, the entropy-based measures are not defined with respect to a baseline quantity (as is the case with the M-ratio). This is why they do not comply with the terms hyper or hypo-sensitivity. But in practice they produce corresponding values for all the M-ratio range, including cases in which the rater is hyper-sensitive. This can be seen in Figures 3, 5 and 6. For example, the heatmaps in Figure 3 record the entire range in which meta-d’>d’. Similarly, in Figures 5 and 6, some participants exhibit M-ratio values that are greater than 1, still the information-theoretic values correspond well with those values.
- The M-Ratios used for empirical validation (Figures 5 & 6) contain M-Ratios in a range that I have never seen before. For the Kent Face Matching Task, the vast majority of observers have M-Ratios exceeding 1 indicating strong to very strong hypersensitivity. I can imagine different technical reasons -- use of a staircase (doi.org/10.1093/nc/niz009), low performance (as d' is in denominator and -- given it's binary outcomes -- is typically noisy), or generally too little trials. If the nature of the task plausibly explains these extreme values, it should be explained. If not, my advice would be to provide a comparison with hierarchically estimated M-ratios (github.com/metacoglab/HMeta-d), possibly replacing meta-d'-d' (see 6.)
I thank the Reviewer for this comment. I now address this issue in the text. The range of the M-ratio values results from the relatively small number of trials in the KFMT.
Minor comments:
- I'm wondering which model the author used to generate data with the meta − d′ model. In particular, how exactly did the author vary "the values of [...] meta − d′ incrementally in small steps (p. 7)" and how did they compute confidence ratings from this? I realize that this sentence refers to another manuscript, but it would be important to know the metacognitive model underlying Figure 3. As a side note, it has only been recently shown that the underlying model of meta-d'/d' (i.e. M-ratio) is of a truncated Gaussian type (doi.org/10.1037/met0000634).
I thank the Reviewer for asking this clarification. I now provide a detailed report on the simulations. I have also created an OSF file with all the Matlab codes, including the ones in which I simulate these data.
- I think there is an error in Figure 6A in that it accidentally shows Meta-J and not Meta-U (Meta-J is also shown in Figure 6C).
The Reviewer is correct. My apologize. I now corrected this mistake.
- I advice to not include analyses related to meta-d - d' in the figures (5D-F, 6D-F) and the text. The measure of meta-d - d' is conceptually flawed and thus only adds noise and no meaningful information. The flaw can be seen when e.g. considering an observer with very high metacognitive noise, such that meta-d' is essentially 0 irrespective of type 1 performance. In this case meta-d' ≈ d', and thus the strong type 1 performance difference of this measure becomes transparent (see also doi.org/10.1093/nc/niab040, Figures 2C & 3A). M-ratio does not contain this flaw, as meta-d' is in the nominator and observers with zero metacognitive introspection will have an M-ratio of 0 regardless of type 1 performance. Thus, I suggest focusing on M-ratio.
I thank the Reviewer for making this important comment. I fully respect their line of argument. I have now removed the meta d – d analyses, and focused on the M-ratio analyses.
Signed
Matthias Guggenmos
I thank the Reviewer for their insightful comments and suggestions.
Sincerely
Daniel Fitousi

Reviewer 2 Report
Comments and Suggestions for Authors
Please see attached pdf file for comments.

Author Response
Reviewer 2
The manuscript tests the convergence of information-theoretic measures of metacognitive efficiency using real empirical data sets. The results support the viability of these measures. Below are my comments.
I thank the Reviewer for their positive evaluation.
1) Lines 59-79 on Page 2: Figure 1 is cited but the authors don’t explain it well. A clearer breakdown explanation of each panel of figure 1 would help the people to understand how this figure relates to the study’s objectives.
I thank the Reviewer for this important comment. I have now added an explanation of Figure 1 that is incorporated in the text.
2) Lines 107-116 on Page 3: Figure 2 is cited but the authors didn’t provide the details of Venn diagram, as well as why this diagram is used in this paper.
I now provide an explanation of this Figure.
3) The authors may want to include a “method” section where (a) they can include The theoretical derivations from Sections 2.1 and 2.2; (b) specify the experimental design and data collection procedures; (c) The numerical methods used in simulations and analyses; (d) specify the software and key parameters used in generating Figures 3, 5, and 6, such as the number of iterations, sampling distributions, and step sizes for d′ and meta-d′; (e) Nowadays, the authors often share the codes to further facilitate future research and replication.
This is a great suggestion. I have now added a Method section for the Experiments. As for the simulations, I have added a paragraph that explains in detail how they were run. In addition, I have created an OSF website that consists of all the code and data. All data, codes, and supporting materials can be downloaded from https://osf.io/hsvfm/
4) Sections 2.1 and 2.2 contain dense mathematical explanations which could be difficult for some readers. I would suggest breaking down key equations with intuitive explanations to indicate their meaning.
I apologize for this inconvenience for the reader. The first Equation by Dayan must be presented as a series of equations and cannot be broke down. The other equations are spaced out and are introduced with intuitive explanations as elaborated as possible.
5) Lines 252-258 on page 8 seems to be a part of introduction section as it lists the important questions that this manuscript addresses and the hypothesis of this study.
I agree. I now added the information that appears in these lines to the first paragraph of the introduction.
6) Studies involving human participants typically require approval from a university ethics board or Institutional Review Board (IRB) before data collection. But this paper does not mention if such approval was obtained.
An approval was obtained from the Ethical committee of Ariel University. I have added details about this approval to the Method section.
7) This manuscript uses face-matching tasks to validate the proposed measures. However, it does not justify why this specific task is well-suited for such validation. For example, the authors may want to briefly discuss (a) why was the face-matching task chosen over other cognitive tasks? (b) how well does this task generalize to other domains of metacognitive efficiency?
The Reviewer is correct. I have now added a paragraph that addresses these points.
“The face-matching task is ideal for our current purposes. It is challenging enough to produce below-ceiling accuracy for all participants [23], enabling us to compute SDT and associated measures. Furthermore, it does not require an additional staircase procedure, which is known to inflate both model-based and model-free measures of metacognitive sensitivity [65]. Moreover, the task is implemented in various well-established face-matching tests [56,66], for which the normative Type 1 performance characteristics are well known. A signal-detection model of the task has been recently advanced [23]. However, little is known about the metacognitive efficiency of observers in this task. Taken together, these factors make the face-matching task a compelling testing platform.”
8) This manuscript supports the convergence of information-theoretic measures using real data sets. However, the limitations of these measures are unclear. For example, (a) does convergence mean these measures can be used interchangeably? (b) are there scenarios where one measure is preferable over another? (c) how do these results inform the design of future metacognition studies? (d) whether one measure is preferable over the others or if they should be used in combination.
Good points. The following paragraph addresses these issues and was added to the General Discussion.
The convergence of the information-theoretic measures suggests that they can be used interchangeably. However, in practical applications, the meta – J, for example, is preferable to the meta -KL for two reasons. First, it is a metric as it complies with the triangle equality (Jeffrey, 1992), and second it takes into consideration reduction in entropy from both the correct and incorrect accuracy-confidence distributions. The meta-KL measure in contrast, is not a metric (Cover & Thomas, 1991), and it also does not consider the reduction in entropy in the incorrect accuracy-confidence distribution. This is important if one is interested in dissociating two alternative mechanisms by which raters weight their evidence. Moreover, the meta-U and meta-I’ are of broader applicability than meta-KL and meta-J, mainly because they are not contingent on the accuracy of the actor’s response. As such, they can be applied to tasks for which the correctness of the actor’s response is not defined (Thurstone, 1927). Another prominent advantage of all the information-theoretic measures is that they can be computed for tasks that require categorization or classification of multi-attribute stimuli (Li, & Ma, 2020). This is currently impossible to achieve with the meta- d’ approach.
Another relevant issue that emerges from the application of information theory to metacognition concerns the granularity of the confidence judgment scale. If the raters use every point in their scale equally often, then a rater that operates with an 8-point scale can be more sensitive than a rater who operates with a 4-point scale. The reason is that the former rater can transmit more information than the latter (Garner, 1960). Both Dayan (2023), and Fitousi (2025) have shown by simulations that the information-theoretic measures accommodate the granularity of the confidence scale, that is, they increase as the number of rating alternatives increases. SDT-related measures do not exhibit this important characteristic. Moreover, the information-theoretic measures accommodate the granularity of the stimuli and response alternatives. Finally, I would recommend researchers to deploy all three major information-theoretic measures together, if possible, to enhance the validity of their conclusions.
9) In the conclusion section, it would be better to have a short discussion on some questions, for example, (a) does this study's rely on face-matching tasks limit generalizability to other metacognitive domains? (b) are there any assumptions in the information-theoretic framework that could introduce biases? (c) are their findings robust to different stimulus types or experimental conditions?
I thank the Reviewer for their suggestion. I have now incorporated several new sections in the General Discussion that address sundry aspects of the information approach.
10) This manuscript has some typographical errors, for example "meatcognition" should be "metacognition". The authors may want to proofread before resubmission.
Thank you. This was correct.
11) Figure 1: Increase the font size of labels in Panels A and B for better readability.
I increased the entire Figure’s size.
12) Figures 5 and 6: specify the meanings of blue circles, red solid lines, and red dashed lines.
This is done now.
I thank the Reviewer for their insightful comments and suggestions.
Sincerely
Daniel Fitousi

Round 2
Reviewer 1 Report
Comments and Suggestions for Authors
The author has convincingly addressed my comments in the revised version of the manuscript.
Reviewer 2 Report
Comments and Suggestions for Authors
The manuscript is good for publication.